# In Vitro Activity of Farnesol against *Malassezia pachydermatis* Isolates from Otitis Externa Cases in Dogs

**DOI:** 10.3390/ani13071259

**Published:** 2023-04-05

**Authors:** Ifarajimi Rapheal Olabode, Nadezhda Sachivkina, Arfenia Karamyan, Ramziya Mannapova, Olga Kuznetsova, Anna Bobunova, Natallia Zhabo, Marina Avdonina, Regina Gurina

**Affiliations:** 1Department of Veterinary Medicine, Agrarian Technological Institute, Peoples’ Friendship University of Russia (RUDN University), 117198 Moscow, Russia; 2Department of Microbiology V.S. Kiktenko, Institute of Medicine, Peoples’ Friendship University of Russia (RUDN University), 117198 Moscow, Russia; 3Department of Veterinary Medicine, Russian State Agrarian University, Moscow Timiryazev Agricultural Academy, 127434 Moscow, Russia; 4Department of Biochemistry T.T. Berezov, Institute of Medicine, Peoples’ Friendship University of Russia (RUDN University), 117198 Moscow, Russia; 5Department of Foreign Languages, Institute of Medicine, Peoples’ Friendship University of Russia (RUDN University), 117198 Moscow, Russia; 6Department of Linguistics and Intercultural Communication, Moscow State Linguistic University, 119034 Moscow, Russia; 7Department of Technosphere Safety, Peoples’ Friendship University of Russia (RUDN University), 117198 Moscow, Russia

**Keywords:** Farnesol, dogs, *Malassezia pachydermatis*, antifungal susceptibility, quorum-sensing molecules

## Abstract

**Simple Summary:**

Despite the discoveries of new therapeutic antimycotics, the development of drug resistance is still the main clinical challenge in the treatment of mycoses. Data on the presence of new phytopreparations, along with the direct fungicidal effects that interfere with the resistance of fungal pathogens located in the biofilm, are of great interest. Our research focused on four investigations: (1) study *M. pachydermatis* as a biofilm infection; (2) the search for the strongest biofilm producers and the most resistant strains to antimycotics; (3) check if there is a correlation between the resistance of strains to modern antimycotics and the ability to form biofilms; and (4) study of the effect on three of the best biofilm producers under the action of several concentrations of Farnesol. These experiments confirmed related applications of Farnesol and its research progression in antifungal therapy of otitis externa.

**Abstract:**

Chronic otitis externa of dogs is a significant problem due to the prevalence and complexity of the treatment of such animals. There is evidence that in 60–80% of cases of infectious diseases microorganisms located in the biofilm phenotype play the main role. Microorganisms in the biofilm phenotype have a number of advantages, the most significant of which is considered to be increased resistance to various external factors. Among them, a special place is occupied by resistance to antibiotics. In recent decades, research has been conducted at an increasing scale on the role of biofilm infections in various pathologies in veterinary medicine. The etiology and therapy of dog otitis externa caused by *Malassezia pachydermatis* biofilm has not been fully studied. This is why we consider relevant the scientific and practical aspects of research on the etiology and therapy of dog otitis externa from the position of biofilm infection. In this work, it has been statistically proven that there is a relationship between the optical density of *Malassezia pachydermatis* biofilms and their sensitivity to drugs, and this relationship is statistically significant. In addition, we have demonstrated that Farnesol has a good antibiofilm effect at a concentration of more 1.6 μM/mL (24% OD decrease of biofilm), and its highest antibiofilm effect (71–55%—more than a half) was observed at a concentration of 200–12.5 μM/mL.

## 1. Introduction

The basidiomycetous yeast (BY) *Malassezia pachydermatis* is a commensal but is also the most isolated pathogenic yeast in veterinary medicine [1,2]. The mechanisms of the transition of *M. pachydermatis* from a commensal to a pathogen are not fully understood. Generally, treatment of *Malassezia*-associated diseases is based on topical application of an antifungal drug combined with antibiotics to control bacterial infection and glucocorticoids to reduce inflammation [3,4,5].

Inflammatory processes of the middle ear, complicated by a fungal infection, are a frequently diagnosed pathology. According to literature data, otitis externa account for 20% of all diseases encountered in veterinary practice. It has also been found that otitis externa of the outer ear in dogs and cats are five times more common than in other animal species [6,7,8].

Currently, there is a tendency to increase the incidence of opportunistic mycoses in animals. Dermatomycoses (*Microsporia, trichophytia*) are replaced by diseases caused by opportunistic fungi. One of these are yeast-like fungi (YLF) of the genera *Candida* and BY *Malassezia* [9,10,11].

Although fungi of the genus *Malassezia* are the most common etiological agents of infectious animal otitis externa, it must be borne in mind that these diseases can also be caused by other types of fungi and yeast. This indicates the need for qualified species identification of microorganism isolated from animals suspected of having *Malassezia* infection.

The pathogenic properties of BY of the genus *Malassezia* and their clinical role in the infectious diseases of animals are still topics of discussion. The factors relating to the transition of a microorganism from a non-pathogenic to a clearly pathogenic form capable of causing a disease has not been fully clarified. There is no consensus on whether *Malassezia* infections can be considered as an independent disease, or whether they are only an aggravating factor against the background of other pathologies [12,13,14].

In veterinary mycology, the clinical role of BY *Malassezia* in animal skin diseases has only been established relatively recently. Intensive research in recent decades has made a significant contribution to the understanding of the pathogenesis of *Malassezia* infections. Currently, most researchers share the opinion that the pathogenicity of *Malassezia* is “opportunistic” in nature, i.e., BY is able to exhibit pathogenic properties only against a background of certain predisposing factors. Therefore, in favorable conditions (increased sebum secretion and humidity, violation of the epidermal barrier), they actively multiply, the yeast form of the fungus turns into mycelial, and *Malassesia* is introduced into the epidermis, showing pathogenic properties. However, according to another theory, the transformation from the yeast phase to the mycelial phase is not due to the special pathogenicity of the latter but is only a consequence of lipid metabolism disorders in the host body. This is based on the fact that the *M. pachydermatis* species is not able to transform into a mycelial form [15,16].

The development of the pathological process in *Malassesiosis* is associated with a multiple increase in the population of microorganisms in the lesion. The population of BY in sick animals increases by 100–10,000 times. Moreover, an increase in the number of cells of the genus *Malassezia* is noted, not only on the surface of the skin, but also on the mucous membranes of the nasal cavity, vulva, and prepuce, i.e., the factors predisposing to this are systemic [11,12,17].

On the one hand, the primary factor for increasing the BY population is a violation of the physical, chemical, and immunological mechanisms of host defense, which normally limit the fungal colonization of the skin [18,19]. On the other hand, communication of microorganisms within the population, carried out by means of “signaling molecules”, plays a key role in starting the pathological process. It is assumed that when a population reaches a certain number, a “sense of quorum” arises in it, which serves as a starting signal for the activation of pathogenicity factors and, as a consequence, leads to the development of an infectious process [20].

The mechanism of intrapopulation communication of microorganisms is associated with such a phenomenon as the formation of biofilms—supra-organizational structures that provide protective and trophic functions. Biofilms are differentiated communities of microorganisms formed by a single microbial agent or a mixture of fungal and bacterial species. Biofilms are attached to biotic or abiotic surfaces, and their structure contributes to the innate physical and chemical resistance of microorganisms [21]. It is known that the ability to form biofilms is one of the pathogenicity factors of the *Candida* genus [22], and in 2007 it was found that BY of the genus *Malassezia* are also capable of forming biofilms on the surface of various substrates [23].

Researchers have found that the symbiotic relationships of BY and skin-dwelling bacteria (in particular, *Staphylococci*) play an important role in the pathogenesis of the disease. *Staphylococci* also produce lipase, which disrupts the secretory function of the skin and creates favorable conditions for the growth of both organisms, while such conditions are unfavorable for other competitive microorganisms [24]. In addition to *Staphylococci*, other types of bacteria and microscopic fungi can play a role in the pathogenesis of the disease. Therefore, from dogs with external otitis caused by *M. pachydermatis* were also isolated bacterias such as: *Staphylococcus* spp., *Pseudomonas* spp., *Proteus* spp., and *Streptococcus* spp., as well as fungi—*Candida* spp. and *Aspergillus* spp. [25,26].

Significant virulence factors of BY *Malassezia* are hydrolytic enzymes that cause the invasion of the fungus into host tissues. Lipolytic enzymes are able to hydrolyze skin secretion lipids to free fatty acids. In turn, free fatty acids inhibit the growth of other microorganisms, increasing the competitiveness of BY [27].

Recently, in the scientific literature, close attention has been paid to the study of herbal medicines and the possibility of their use for the treatment of infectious (fungal and bacterial) diseases [28]. According to the literature and our own research, Farnesol (C_15_H_26_O)—Far—has proven its antimycotic efficacy in in vivo and in vitro models [29,30,31]. Chemically, Far is an acyclic sesquiterpene alcohol; it is a thermally stable molecule. It is not exposed to extreme pH values, which is especially important in the development of YLF infection. Farnesol, as a quorum-sensing (QS) molecule, participates in the regulation of various physiological processes in unicellular fungi, including filamentation, biofilm formation, drug susceptibility, and apoptosis. This compound is produced by many microorganisms and is also contained in various essential oils of plants, for example, in the flowers of *Tilia europaea* [32]. Despite a large number of scientific articles and our own studies proving the antibiotic effect of Far against *Candida* spp., the effect of this molecule in relation to BY of the genus *Malassezia* has not yet been studied.

Today, in veterinary medicine, only a few doctors recommend phytopreparations for the treatment of otitis externa and dermatitis, and there is still relatively little data on their effectiveness in the scientific literature, especially in Russian. Therefore, the development of targeted therapy using alternative means can become one of the directions in solving global problems of infectious animal diseases, as well as the increased resistance of microorganisms [10,17,33]. Essential oils of herbal origin, such as tea tree oil, lime oil, and rosemary oil, have gained global importance in dermatology. These oils are rich in aromatic secondary metabolites, especially terpenes and phenolic components that impart substantial antimicrobial properties and resisting biofilm production [34,35,36,37,38].

Our research focused on four investigations: (1) study *M. pachydermatis* as a biofilm infection; (2) the search for the strongest biofilm producers and the most resistant strains to antimycotics; (3) check if there is a correlation between the resistance of strains to modern antimycotics and the ability to form biofilms; and (4) study of the effect on three of the best biofilm producers under the action of several concentrations of Far. These experiments confirmed related applications of Far and its research progression in antifungal therapy of otitis externa.

## 2. Materials and Methods

### 2.1. Animals

At the Center of Veterinary Innovative Medicine of the RUDN (Moscow) for the period 2021–2022, 76 cases of dog otitis externa were investigated, of which 30 cases were confirmed otitis with *Malassezia pachydermatis* etiology (C1–C30).

The study involved 30 dogs of different breeds, gender, and age, from 1 to 14 years. All the animals had apartment maintenance with walking. Their diets consisted of dry food. Treatments for ectoparasites (external) and endoparasites (internal) were carried out in all participants of the experiment regularly and on time. According to their anamnesis, there was itching in the ears and an unpleasant smell for several weeks. At the reception, the following were noted: hyperemia of the auricles, stenosis of the auditory canal, in some cases alopecia, and a large amount of yellow-brown discharge with a sharp sour smell (Figure 1a–f). We did not conduct any animal experiments; all this work was only with microbial strains and did not require authorization from the Ethics Committee.

In our work, several clinical forms of external otitis were noted. Erythematous otitis is accompanied by erythema and edema, and the degree of itching varied. Erythematous-ceruminous was manifested by erythema, itching, and the release of abundant ear secretions (cerumen) of a yellow-brown color, often with an unpleasant odor. Ceruminous otitis was characterized by abundant discharge of earwax, but without signs of inflammation. The proliferative form was characterized by hyperplasia of the sebaceous glands, the formation of papules merging in the form of “cauliflower”, which is typical in chronic cases. With purulent otitis, purulent discharge was observed, palpation of the ear was painful, and sometimes crepitation was heard. In most cases, we observed *Malassezia*-otitis with erythematous-ceruminous form.

### 2.2. Strains

For our experiments, BY received clinically was used. Preliminary identification of strains to the genus *Malassezia* level was carried out by phenotypic signs using microscopy of an ear smear contents and staining with Gentian violet. Ear exudate was applied to the surface of the nutrient medium Sabouraud dextrose agar (Difco, Bordeaux, France) and cultured at 37 °C for 48–72 h. Then identification of microorganisms was carried out using the matrix-activated laser desorption/ionization technology “Bruker Daltonik MALDI Biotyper” (“Bruker Daltonik Inc.”, Bremen, Germany). Identification with a Score of more than 2000 was considered reliable. The obtained spectra were compared with the library of mass spectrum profiles of Biotiper3 MALDI [39]. The strains were stored at −80 °C [40].

### 2.3. Reagents

Farnesol (farnesol) (trans, trans-farnesol; Sigma-Adrich, Darmstadt, Germany), molar mass = 222.37 g/mol, mass of the substance = 0.886 g/mL, the amount of the substance in moles = 0.886:222.37 = 0.004 M/mL, or 4000 µM/mL (Figure 2).

For *M. pachydermatis* cultivation, Sabouraud dextrose agar (SDA) or Sabouraud dextrose broth (SDB) were used (Difco, Bordeaux, France). BY of the genus *Malassezia* have a special metabolism; therefore, their growth requires the presence of fatty acids in the nutrient medium with a chain length of 12–14 carbon atoms. They are not able to synthesize such fatty acids on their own, except for *M. pachydermatis*, which is able to grow on simple nutrient media such as SDA and SDB.

### 2.4. Densitometric Indicators of Microbial Biofilms

Colonies of four-day cultures of *M. pachydermatis* from SDA were washed with physiological solution (PhS) (pH 7.0). The concentration of BY was 0.5, according to McFarland, which corresponded to 1.5 × 10^8^ cells/mL. The tested samples were added to the wells of a 96-well plate (Medpolymer Company, Russia), cultivated in a constant aerobic environment at 37 °C; for 72 h. The liquid was discarded, and the wells were washed with 200 μL of phosphate-buffered solution (PBS) three times (pH 7.3). The plates were shaken for 5 min at each stage of washing. The samples were fixed with 150 μL of 96% ethanol for 15 min and dried at 37 °C for 20 min. The microbial biofilms were stained by adding 0.5% stain solution—crystal violet (HiMedia™ Laboratories Pvt. Ltd., Mumbai, India)—in each well and subsequent cultivation at 37 °C for 5 min. The optical density (OD) of the biofilm was measured by the degree of binding of crystal violet at a wavelength of 580 nm (OD_580_) in an Immunochem-2100 microplate photometric analyzer (HTI, North Attleboro, MA, USA) [28,30]. That is, for weak biofilm producers, the optical density of the sample or the culture of microorganisms (density of the sample, ODs) is less than two times (ODs ≤ 0.194) greater than the optical density of the control, i.e., nutrient medium without inoculum (density of the control, Dc); the optical density of the sample exceeds the optical density of the control by 2–4 times (ODs = 0.194–0.388) for moderate or average biofilm producers; the optical density of the sample exceeds the optical density of the control by more than four times (ODs ≥ 0.388) for strong biofilm producers. Density of control (Dc) was 0.097 ± 0.005. For statistical analysis (Chi-square), the optical density (factorial sign) indicators were converted into numbers—degree OD. A weak biofilm producer was given 1 point, average—2 points, and strong—3.

### 2.5. Preparation of M. pachydermatis Cultures and Assessment of Their Susceptibility to Antimycotics

The sensitivity of BY *M. pachydermatis* to antimycotic drugs was investigated using the Kirby–Bauer’s disk diffusion method, exactly as described in our previous study [41,42,43]. The antimycotics tested were nystatin (NS; 50 µg), amphotericin B (AP; 10 µg), ketoconazole (KT; 10 µg), clotrimazole (CC; 10 µg), voriconazole (VOR; 10 µg), fluconazole (FU; 25 µg), miconazole (MIC; 10 µg), and intraconazole (IT; 10 µg) (HiMedia™ Laboratories Pvt. Ltd., Mumbai, India). All strains recovered from animals with otitis externa were classified as susceptible (S)—1 point, intermediate (I)—2 points, or resistant (R)—3 points, according to the manufacturer’s breakpoints for yeasts. These scores were entered into a table and calculated as Degree of resistance (Dr) to antifungal drugs (resulting sign) for the statistical analysis Chi-square test.

### 2.6. M. pachydermatis Biofilms Processing with Farnesol

A method for studying the influence of different concentrations of Far on YLF biofilms, especially on *Candida* spp., is described in our previous studies [28,30]. The same method was used here without any modification. Briefly, the automatic pipette was inserted into the wells of a 96-well plate (Medpolymer, St. Petersburg, Russia):An amount of 100 µL SDB in each of the 12 holes of the first row, A.An amount of 100 µL of Far was added at an initial concentration of 400 µM to the second well of the first row. The first hole was left as a control. In the second hole, the volume was 200 µL, and the concentration of Farnesol was 200 µM. By successive transfer of 100 µL of the solution from the second well to the third, from the third to the fourth…, etc., the concentration of Far was reduced by half each time.Then, in each well of the first row, starting from the first, 100 µL of *M. pachydermatis* culture in SDB was added at a concentration of 4 units (McFarland).

An overview of the sequence of stages of the study into the effect of Far on the formation of *M. pachydermatis* biofilms is presented in Table 1.

The total volume of the wells was 200 µL. The experiment was repeated three times, 3 µL plates were used, and four rows were used per plate. The microliter plates were cultured with the lid closed at 37 °C for 72 h.

The average decrease was measured and used to calculate the biofilm inhibition percentage by Far, where OD AS is the optical density average of the samples of *M. pachydermatis* in the experiment, and OD AC is the optical density average of isolates in the control without Far:(1)$$\;Average\;decrease\;OD\left(\%\right)=\frac{OD\;AS\;x\;100}{OD\;AC}-100$$

### 2.7. Statistical Analysis

All the results were expressed as mean  ±  standard error of mean (SEM) of at least three replicates. The criterion Chi-square was used for the analysis of the contingency tables. Such a table is a good means of representing the joint distribution of two variables, and is designed to explore the relationship between them. The cross table is the most versatile tool for studying statistical relationships, as it can represent variables with any level of measurement. The lines of the contingency table correspond to the values of one variable—optical density, the columns—to the values of another variable—the resistance of microorganisms. The data for these two criteria were preliminarily grouped into intervals. The statistical significance was set at *p* ≤ 0.05, and where applicable, the difference between samples was assessed using the statistical software XLSTAT 2020 (Addinsof Inc., New York, NY, USA). The effect of different concentrations of Far on the degree of biofilm formation was evaluated with a logarithmic-logistic distribution. This model was chosen because the Cox model’s proportional hazard assumption did not fit all covariates. All the graphs were plotted using Microsoft Excel (Microsoft Excel for Office 365 MSO, Microsoft COP., Redmond, WA, USA).

## 3. Results

### 3.1. Densitometric Indicators of Microbial Biofilms and Their Susceptibility to Antimycotics

Densitometric indicators of the sample are shown in Table 2. According to our gradation, 2 (6.7%) strains belong to weak producers of biofilms, 25 (83.3%) strains belong to moderate producers, and 3 (10%) to strong ones. It is worth emphasizing the great importance of belonging to a high index of biofilm formation in our clinical strains of *M. pachydermatis*. A moderate level of biofilm formation indicates a good adhesion of these opportunistic fungi. Our team has experience in determining the degree of biofilm formation in clinical *Candida* spp. Compared to the genus *Malassezia*, the genus *Candida* can reach a higher level in biofilm production [20,22,28,30]. Analyzing the densitometric indicators, it was found that the studied *Malassezia pachydermatis* C23, C27, and C3 are the strongest producers of biofilms.

The isolates were reported as sensitive, intermediate, and resistant in general, according to the manufacturer using the CLSI guidelines’ breakpoints for yeasts [12]; most strains were sensitive (S) to antimycotics drugs. However, a few exceptions were observed: three strains were classified as resistant (R) to AP, MIC, and VOR; four were R to KT; seven were R to FU; and only one was R to IT. There were no R results for NS and CC antifungal discs. The strain *M. pachydermatis* C23 was classified as R to AP, KT, and FU; I to NS, CC, IT, and VOR; and S only to MIC. Strain *M. pachydermatis* C27 was classified as R to VOR; I to AP and KT; and S to others. Strain *M. pachydermatis* C3 was classified as R to FU and I to CC, VOR, and MIC.

It was very interesting to see if there was a relationship between the ability to form biofilms and resistance to the antifungals tested. One possibility to find out was to calculate a resistance score, assigning a value of 1 if the isolate was sensitive, a value of 2—intermediate, and a value of 3—resistant. In this way, a numerical value was obtained for each isolate, and we have adopted rules, such as that a yeast isolate is considered sensitive if the sum of the scores for all eight antimycotic drugs is 8–10, intermediate—11–13, and resistant—14–18. In addition, three categories have been defined for the ability to form biofilms (weak—1, moderate—2, and strong—3). A table was constructed, and statistical analysis was performed (Chi-square) (Table 3).

Calculation results: The value of the χ2 criterion is 10.421 (sum of all points, Chi-squared). The critical value of χ2 at the significance level *p* = 0.05 is 9.488. The relationship between the factorial and performance characteristics is statistically significant at a significance level of *p* < 0.05 (the Chi-squared value is greater than the critical value). Significance level *p* = 0.034.

We combined all our observations (30 isolates) into nine groups, depending on the optical density and the degree of resistance to eight antifungal drugs:1st group: weak biofilm producer—sensitive to antifungal drugs: *Malassezia pachydermatis* C13;2nd group: weak biofilm producer—intermediate resistance to antifungal drugs: *Malassezia pachydermatis* C9;3rd group: weak biofilm producer—resistant to antifungal drugs: no isolates;4th group: moderate biofilm producer—sensitive to antifungal drugs: *Malassezia pachydermatis* C6, 7, 8, 12, 14, 16, 18, 22, 24, 26, 28;5th group: moderate biofilm producer—intermediate resistance to antifungal drugs: *Malassezia pachydermatis* C1, 2, 4, 5, 10, 11, 15, 17, 19, 20, 21, 25, 29, 30;6th group: moderate biofilm producer—resistant to antifungal drugs: no isolates;7th group: strong biofilm producer—sensitive to antifungal drugs: no isolates;8th group: strong biofilm producer—intermediate resistance to antifungal drugs: *Malassezia pachydermatis* C3, 27;9th group: strong biofilm producer—resistant to antifungal drugs: *Malassezia pachydermatis* C23.

It has been statistically proven that there is a relationship between the optical density of *Malassezia pachydermatis* biofilms and their sensitivity to drugs, and this relationship is statistically significant. Analyzing the densitometric indicators, it was found that the studied C23, C27, and C3 are the strongest producers of biofilms. In addition, they have an intermediate and resistant score to antifungal drugs. Therefore, it was decided to continue the experiment with this particular isolates of BY.

### 3.2. BY Malassezia pachydermatis Biofilm Inhibition by Farnesol

The average decrease was measured and used to calculate the biofilm inhibition percentage by Far, where *OD AS* is the optical density average of thee isolates (*M. pachydermatis* C23, 27, and 3) in the experiment and *OD AC* is the optical density average of thee isolates (*M. pachydermatis* C23, 27, and 3) in the control without Far (Table 4).

The Far demonstrated good antibiofilm effects at a concentration of more than 1.6 μM/mL (24% OD decrease of biofilm) (Figure 3), and its highest antibiofilm effects (71–55%—more than a half) were observed at a concentration of 200–12.5 μM/mL. The non-linear logarithmic-logistic regression model was used to construct a dose-response curve for Far antifungal effect.

## 4. Discussion

The increasing dominance of polyresistant strains of *Malassezia* spp. has forced scientists worldwide to consider new approaches for antimycotic drugs. A prerequisite is the safety and non-toxicity of new drugs for the patient [1,6,21]. Accumulation of *M. pachydermatis* on the animal skin surface can lead to subsequent invasion, which can cause the lymphogenic and hematogenic spread of fungi, generalization of the process, and dissemination into internal organs [24,29].

Classical methods remain the main ones in the etiological diagnosis of mycoses, such as cultivation of microorganisms on nutrient media, microscopy of biomaterial, and isolation of pure culture of the pathogen with its subsequent identification.

Algorithms for the laboratory diagnostics of animal fungal and yeast infections have changed significantly in recent years. New tests based on molecular genetic methods (PCR) have appeared, and proteomic methods for the identification of microorganisms using matrix-activated laser desorption-ionization time-of-flight mass spectrometry (MALDI-TOF MS) for direct profiling of bacteria and fungi have been developed, which are being introduced into the work of microbiological laboratories and has revolutionized the field of the species identification of microorganisms.

In addition to PCR and the sequencing of species-specific DNA loci, other molecular genetic approaches can be used for species identification of *Malassezia furfur*, such as random polymorphic DNA amplification (RAPD), restriction analysis of PCR amplicons of ribosomal sequences, analysis of nucleotide DNA sequences, gel electrophoresis in a pulsating field (PFGE), polymorphism of the length of amplified fragments (AFLP), and denaturing gradient gel electrophoresis (DGGE) [39,44,45,46]. In some cases, such methods can be useful for species identification of *Malassezia pachydermatis*, effectively distinguishing this species from other species of the genus *Malassezia*, though they are not yet suitable for active use in routine laboratory practice [47].

The sensitivity of fungi and yeast to antifungal drugs depends on their species. In this regard, constant monitoring of the species composition of fungi is necessary. The choice of antifungal drugs depends on the type of pathogen, the characteristics of pharmacokinetics, and the clinical condition of the patient.

To determine the sensitivity of *M. pachydermatis* to antifungal drugs, two main methods are used: the method of serial dilutions in a liquid or solid nutrient medium and the disco-diffusion method [48,49]. There are two organizations in the world that are developing methods for determining the sensitivity of microorganisms to antimicrobial drugs, as well as criteria for interpreting the results obtained: CLSI (Clinical Laboratory Standards Institute) in the USA and EUCAST (European Committee on Antimicrobial Susceptibility Testing) in Europe. The latest version of the EUCAST recommendations, including recommendations on mushrooms, is the version published in October 2021. It is important to note that in these recommendations there is no information on methods for determining the sensitivity of *M. pachydermatis* to antimycotics, and there are no criteria for interpreting the results obtained [50,51,52].

The results of this study indicate that some *M. pachydermatis* strains can be resistant to same antifungals. At the same time, the other investigators [53,54] support the hypothesis that the treatment of *Malassezia* biofilm infections requires higher drug concentrations than those currently used. Therefore, in the chronic form of *Malassezia* infections, it is necessary to find out the susceptibility of the isolated yeast to the antifungal agents, which will be used for the treatment [55,56], or use additional treatment such as Far.

Much research has been devoted to studying the role of Far as an inhibitor of hyphal morphogenesis [41,43]; however, little research has been published regarding the impacts of Far on fungal virulence and the development of *M. pachydermatis* biofilms.

Far affected the formation of biofilms to a greater extent than planktonic cells. However, despite intensive research on Farnesol over the last decade, how *C. albicans* cells sense Far or how this QS molecule exerts its biological effects remains unclear [30,57].

Furthermore, Farnesol and its derivatives have exhibited antibiofilm, anticancer, antitumor, and fungicidal properties [12,21,29]. Thus, another positive effect of Far in treatment could be its additional antibacterial effect. This was already shown in vitro against several bacterial veterinary isolates [34,36,48,55,56,57,58,59]. In addition, Far can exhibit anti-inflammatory potential [57,60,61,62], which might be another positive attribute for treatment of otitis externa.

The antibiofilm activity of Far has been described according to the time of administration during the development of a *Candida* biofilm as well as the concentration used. However, to date, the effect of Far on BY *Malassezia pachydermatis* has not yet been studied at all. In our experiment, we proved that there is a correlation between the resistance of *M. pachydermatis* strains to modern antimycotics and the ability to form biofilms, and we found that Far affected the biofilm formation degree of BY, which supports its potential as an antimicrobial agent.

## 5. Conclusions

In summary, our results suggested that Farnesol perfectly destroys the biofilms of *M. pachydermatis* in vitro and may be promising for its inclusion in the therapy of dog external otitis. In our study, we have shown that *M. pachydermatis* strains (93.3%) were biofilm producers with varying levels of intensity: moderate or strong. The Far demonstrated good antibiofilm effects at a concentration of more than 1.6 μM/mL (24% OD decrease of biofilm), and its highest antibiofilm effects (71–55%—more than a half) were observed at a concentration of 200–12.5 μM/mL. Given the increased patient sensitivities to existing medications reported in the literature, reducing pathogenicity and reducing biofilm formation via Farnesol could be significant for the development of future therapies. This experiment may also provide a deeper understanding of the potential antifungal mechanism of QS molecules as we search for new solutions to counter infections involving *M. pachydermatis*. BY isolated from the dogs used in this study showed possible resistance against six antimycotics from a total of eight. Combined with conventional antimicrobial therapies, the therapeutic potential of this QS molecule—Far—on the virulence factors of pathogens such as *M. pachydermatis* should be considered for biofilm-associated diseases.

## Figures and Tables

**Figure 1 animals-13-01259-f001:**
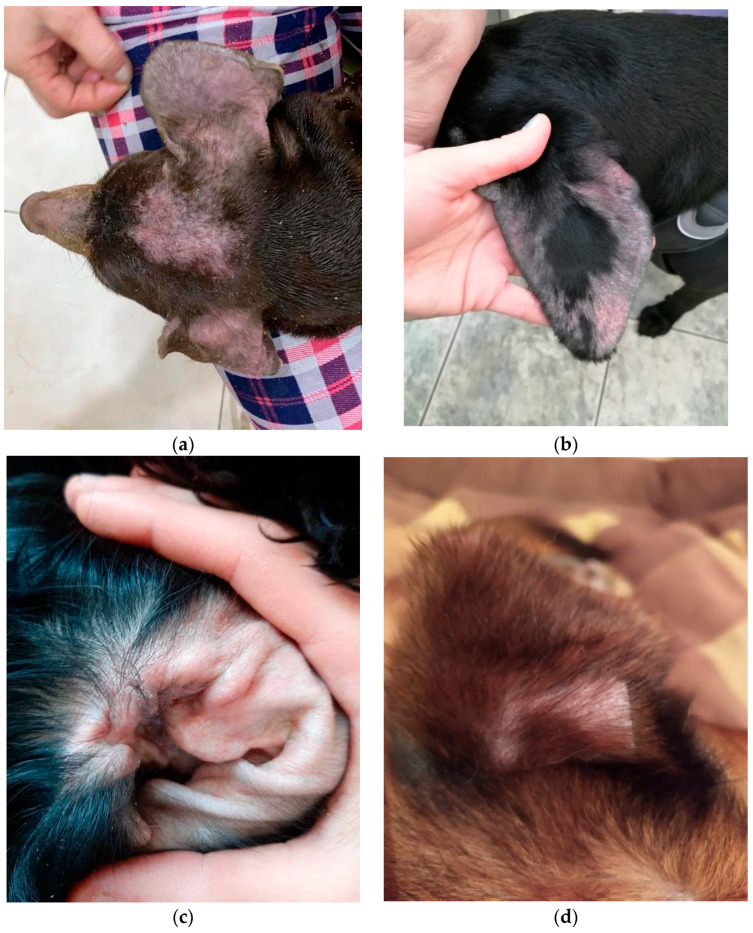
Different clinical forms of dog’s external otitis: (**a**,**b**) erythematous otitis; (**c**,**d**) erythematous-ceruminous; (**e**,**f**) ceruminous.

**Figure 2 animals-13-01259-f002:**
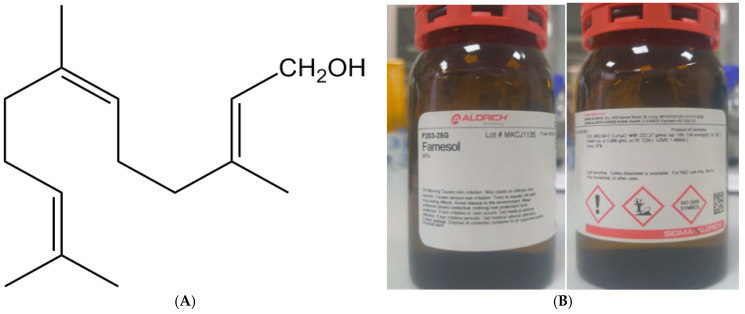
(**A**) the chemical formula of Farnesol; (**B**) the appearance of the drug Farnesol (Far) (Sigma-Aldrich, Darmstadt, Germany).

**Figure 3 animals-13-01259-f003:**
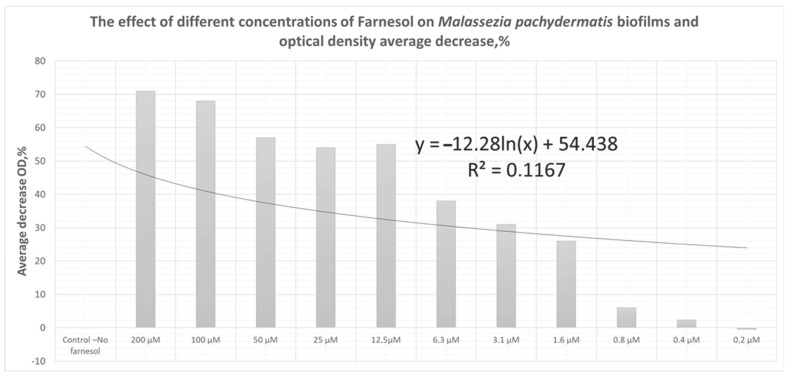
In vitro activity of Farnesol against *M. pachydermatis* (*n* = 3) biofilms and dose-response curve.

**Table 1 animals-13-01259-t001:** Stages of the study of the Farnesol effect on *M. pachydermatis* biofilm formation.

	1	2	3	4	5	6	7	8	9	10	11	12
Action 1	SDB 100 µL	SDB 100 µL	SDB 100 µL	SDB 100 µL	SDB 100 µL	SDB 100 µL	SDB 100 µL	SDB 100 µL	SDB 100 µL	SDB 100 µL	SDB 100 µL	SDB 100 µL
Action 2		+Farnesol										
Action 3	Not titrated	Transfer 100 µL	Transfer 100 µL	Transfer 100 µL	Transfer 100 µL	Transfer 100 µL	Transfer 100 µL	Transfer 100 µL	Transfer 100 µL	Transfer 100 µL	Transfer 100 µL	Transfer 100 µL
The concentration of farnesol	Control– no farnesol	200 μM	100 μM	50 μM	25 μM	12.5 μM	6.3 μM	3.1 μM	1.6 μM	0.8 μM	0.4 μM	0.2 μM
Action 4	+100 µL of culture	+100 µL of culture	+100 µL of culture	+100 µL of culture	+100 µL of culture	+100 µL of culture	+100 µL of culture	+100 µL of culture	+100 µL of culture	+100 µL of culture	+100 µL of culture	+100 µL of culture
Action 5	wait for 72 h											

**Table 2 animals-13-01259-t002:** Determination of the biofilm formation intensity of *Malassezia pachydermatis* stains by optic density and their susceptibility to antimycotics. The richer the color, the higher the corresponding value.

Microorganism	Optic Density	Degree OD	Antimycotic Drugs								Degree of Resistance
			NS	AP	KT	CC	VOR	FU	MIC	IT	
*Malassezia pachydermatis* C1	0.203 ± 0.016	2	1	1	3	1	1	2	1	1	11
*Malassezia pachydermatis* C2	0.351 ± 0.018	2	1	2	1	2	2	3	1	1	13
*Malassezia pachydermatis* C3	0.400 ± 0.012	3	1	1	1	2	2	3	2	1	13
*Malassezia pachydermatis* C4	0.287 ± 0.018	2	1	3	1	2	1	1	1	3	13
*Malassezia pachydermatis* C5	0.261 ± 0.011	2	2	1	1	2	2	1	1	1	11
*Malassezia pachydermatis* C6	0.312 ± 0.029	2	1	1	1	1	1	3	1	1	10
*Malassezia pachydermatis* C7	0.255 ± 0.010	2	1	1	1	1	1	1	1	1	8
*Malassezia pachydermatis* C8	0.243 ± 0.026	2	1	1	1	2	1	1	1	1	9
*Malassezia pachydermatis* C9	0.190 ± 0.016	1	1	3	1	1	1	3	1	2	13
*Malassezia pachydermatis* C10	0.237 ± 0.015	2	2	1	1	1	1	1	2	2	11
*Malassezia pachydermatis* C11	0.345 ± 0.011	2	1	1	3	1	2	2	1	1	12
*Malassezia pachydermatis* C12	0.323 ± 0.017	2	1	2	1	1	2	1	1	1	10
*Malassezia pachydermatis* C13	0.192 ± 0.012	1	1	1	1	2	1	2	1	1	10
*Malassezia pachydermatis* C14	0.258 ± 0.011	2	1	1	1	2	1	1	1	1	9
*Malassezia pachydermatis* C15	0.261 ± 0.010	2	1	1	2	2	1	1	1	2	11
*Malassezia pachydermatis* C16	0.313 ± 0.007	2	1	1	1	1	1	2	1	1	9
*Malassezia pachydermatis* C17	0.294 ± 0.019	2	1	1	3	1	1	3	1	1	12
*Malassezia pachydermatis* C18	0.286 ± 0.020	2	1	1	1	2	1	1	1	1	9
*Malassezia pachydermatis* C19	0.362 ± 0.014	2	2	1	2	2	1	1	1	1	11
*Malassezia pachydermatis* C20	0.366 ± 0.015	2	1	2	1	1	3	1	1	1	11
*Malassezia pachydermatis* C21	0.280 ± 0.016	2	1	2	1	1	1	1	3	1	11
*Malassezia pachydermatis* C22	0.344 ± 0.018	2	1	1	1	1	1	2	1	1	9
*Malassezia pachydermatis* C23	0.441 ± 0.016	3	2	3	3	2	2	3	1	2	18
*Malassezia pachydermatis* C24	0.370 ± 0.015	2	1	1	2	2	1	1	1	1	10
*Malassezia pachydermatis* C25	0.323 ± 0.017	2	2	1	1	1	1	2	3	2	13
*Malassezia pachydermatis* C26	0.288± 0.012	2	1	2	1	1	1	1	1	1	9
*Malassezia pachydermatis* C27	0.403 ± 0.026	3	1	2	2	1	3	1	1	1	12
*Malassezia pachydermatis* C28	0.368 ± 0.014	2	1	1	1	1	1	1	1	1	8
*Malassezia pachydermatis* C29	0.297 ± 0.011	2	1	1	1	1	3	2	1	1	11
*Malassezia pachydermatis* C30	0.353 ± 0.019	2	1	1	2	1	1	3	3	1	13

Note: ODc = 0.097 ± 0.005.

**Table 3 animals-13-01259-t003:** The relationship between the ability to form biofilms and resistance to the antifungals (Chi-square statistical analysis).

Biofilm Optical Density (Factorial Sign)	Degree of Resistance to Antifungal Drugs (Resulting Sign)			
	Sensitive 8–10	Intermediate 11–13	Resistant 14–18	Total
weak—1	1	1	0	2
moderate—2	11	14	0	25
strong—3	0	2	1	3
total	12	17	1	30

**Table 4 animals-13-01259-t004:** The effect of different concentrations of Farnesol on Malassezia pachydermatis biofilms and optical density average decrease,% densitometric studies.

	1	2	3	4	5	6	7	8	9	10	11	12
Far concentration	Control no farnesol	200 μM	100 μM	50 μM	25 μM	12.5 μM	6.3 μM	3.1 μM	1.6 μM	0.8 μM	0.4 μM	0.2 μM
*Malassezia pachydermatis* C23	0.441 ± 0.016	0.120 ± 0.008	0.135 ± 0.011	0.234 ± 0.011	0.226 ± 0.019	0.233 ± 0.010	0.249 ± 0.014	0.302 ± 0.015	0.368 ± 0.017	0.407 ± 0.016	0.439 ± 0.011	0.453 ± 0.012
*Malassezia pachydermatis* C27	0.403 ± 0.026	0.117 ± 0.016	0.121 ± 0.009	0.144 ± 0.013	0.186 ± 0.018	0.272 ± 0.016	0.284 ± 0.011	0.307 ± 0.008	0.320 ± 0.014	0.379 ± 0.012	0.393 ± 0.010	0.399 ± 0.010
*Malassezia pachydermatis* C3	0.400 ± 0.012	0.123 ± 0.018	0.142 ± 0.014	0.160 ± 0.021	0.154 ± 0.017	0.185 ± 0.009	0.234 ± 0.014	0.252 ± 0.012	0.262 ± 0.009	0.279 ± 0.010	0.383 ± 0.014	0.398 ± 0.008
Average OD of 3 isolates	0.415	0.120	0.133	0.179	0.189	0.230	0.256	0.287	0.317	0.390	0.405	0.417
Average decrease OD, %	0	71	68	57	54	55	38	31	24	6	2.4	−0.5

## Data Availability

All data are contained within the article.

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
