# Peer review of "In Vitro Activity of Farnesol against Malassezia pachydermatis Isolates from Otitis Externa Cases in Dogs"

_animals, 2023, doi:10.3390/ani13071259_

Round 1

Reviewer 1 Report (Previous Reviewer 1)

The authors have improved the manuscript. Thank you. 

Author Response

Dear reviewer. Your comments were very valuable to us and we are very grateful to you. Thank you for your opinion.

Reviewer 2 Report (Previous Reviewer 3)

This is a resubmitted manuscript that had previously been reviewed by this reviewer, so either I cannot or the authors have not responded to my comments and suggestions. Although some of the reviewer's comments have been taken into account in this new version, I still consider that the present study does not meet sufficient quality for acceptance in the journal Animals.

Author Response

Dear reviewer. Your comments were very valuable to us and we are very grateful to you. We have taken into account all the corrections. It's a pity that you still think that we can't publish in Animals. We disagree with you. Thank you for your opinion.

Reviewer 3 Report (Previous Reviewer 2)

The requirements indicated has be reevaluated and they substantially improved the quality of the initial article. 

Author Response

Dear reviewer. Your comments were very valuable to us and we are very grateful to you. Thank you for your opinion.

This manuscript is a resubmission of an earlier submission. The following is a list of the peer review reports and author responses from that submission.

Round 1

Reviewer 1 Report

Please edit the manuscript:

1- “targeted targeted” repetitive 

2-“In addition, in addition” repetitive

3-Farnesol (farnesol) (trans, trans-farnesol; Sigma-Adrich, Darmstadt, “Germany”) then Figure 2. B- the appearance of the drug Farnesol (Far) (Sigma-Aldrich, “USA”): Germany or USA? 

4- I recommend that authors make a clear distinction between the otitis externa and “otitis” in general and “Otitis media” in the introduction. This section of the manuscript is confusing. As it appears in the title and all pictures provided this study is about “Otitis externa”. In one place it was mentioned as “otitis media of the outer ear” and there was mention of otitis media on 3 other occasions.

5- The authors tested 3 isolates out of 30: (M. pachydermatis C23,27 and 3). I would recommend that authors clearly explain how these 3 isolates (vs the other 27) were selected to avoid selection bias. This also needs to be mentioned in the conclusion that this is only based on the average result of the 3 selected isolates as the limitation of this study. 

6- The authors have self-cited their previous publications several times. I recommend explaining the methodology in the current manuscript and avoiding self-citations. 

Reviewer 2 Report

The purpose of this work consisted of evaluating the Farnesol on clinical isolated of Malassezia pachydermatis. The theme seems interesting but the article is necessary to make some corrections

1.Abstract:

The abstract describes only general aspects not focusing on the results obtained in the study. I think it is important to add results to increase the quality of the article, also suggesting the novelty aspects.

2.The introduction should focus more on the studied topic. Because the tested substance is a commercial product, the chemical composition of the substance, their possible uses and the results obtained should be mentioned.

3.It is necessary to describe the images, the lesion type. Otherwise, it is hard to follow the text.

4.I don't think it is necessary to add a picture with the tested product

5.In the work at point 2.4 it is mentioned about tested samples, but you did not indicate the concentrations, the method of preparation of the tested substance, the positive, negative control? Thus, the interpretation and understanding of the organizational chart of the experiment is deficient

6.What kind of dye did you use to quantify biofilm? Because the authors mention only 0.5% stain solution?

7. During the assessments have you used control groups?

8. How did you evaluate the biofilm potential?

9. You do not specify how many Malassezia pachydermatis strains you used for the study.

10. It would be interesting if you could add some pictures with the results from the Kirby Bauer test

11. Point 2.6 should be reconsidered as the described methodology is not understood, or could you describe it in the form of a scheme.

12. The evaluation methodology from point 3.2. it should appear in the materials and methods and not in the results.

13. It is necessary to restructure the results

14. The discussions should be more focused on the results obtained in the study compared to other similar studies

Reviewer 3 Report

The manuscript reports only in vitro effects of Farnesol against some M. pachydermatis isolates from otitis externa cases in dogs. Because of its limited innovativeness and methodological unclarities, the reviewer do not recommend the manuscript for publications in Animals. Even so, the results are interesting and the reviewer recommends that the authors substantially improve the quality of the manuscript and consider sending it to another journal for possible publication.

The main weaknesses of the manuscript are the follows:

Q1. Only in vitro results on the effect of farnesol on some isolates of M. pachydermatis from cases of otitis externa in dogs are presented. There is a large amount of information in this regard in the literature on a large number of phytocompounds, including their effect on the formation of biofilms. However, in vivo trials are scarce, so if they were included it would considerably increase the impact of the research.

Q2. Redefine the objectives of the study. In the summary and at the end of the introduction it is mentioned that the study focuses on three investigations, but the reviewer considers that they are not well defined because "M. pachydermatis as a biofilm infection" is not really studied, only some isolates of M. pachydermatis were obtained from otitis externa cases in dogs and its in vitro capacity to produce biofilms was studied. In addition, the ability of farnesol to inhibit the formation of biofilms is not studied, only its fungicidal effect (see Q3).

Q3. The ability of farnesol to inhibit the formation of M. pachydermatis biofilms is not being evaluated; by adding the farnesol before the yeasts only are evaluating its antifungal capacity. In order to evaluate its ability to inhibit biofilms, the yeasts would have to be added first and after incubating for different periods of time, add the farnesol.

Q4. Why is the antifungal effect of farnesol tested only with the three isolates with the greatest capacity for biofilm formation? It could also be interesting to test it with those isolates that showed greater resistance to the tested antifungals.

Q5. In relation to the above, it would be very interesting to see if there is a relationship between the ability to form biofilms and resistance to the antifungals tested. One possibility would be to calculate a resistance score, assigning, for example, a value of 0 if the isolate is sensitive, a value of 1 if it is intermediate, and a value of 2 if it is resistant. In this way, a numerical value would be obtained for each isolate (for example, it would be 3 for isolate C1; 5 for C2 and so on) and since three categories have been defined for the ability to form biofilms (weak, moderate and strong), it could be carried out a table and perform statistical analysis (chi square for example).

Q6. Reformulate the conclusions because the one that appears in the manuscript: "in summary, our results suggested that the farnesol perfectly destroys biofilms of M. pachydermatis and can be used in the therapy of dog otitis" cannot be extracted from the results obtained in the study.

Other considerations:

The reviewer considers that the introduction is too long and does not reflect the state of the art of the topic. Reference is made several times to otitis media when the manuscript is about otitis externa. I think that the term defined as YLF (yeast-like fungi) is not appropriate, yeasts are true fungi, I think the authors are referring to the ability of some yeasts to form mycelium, but in my opinion the term yeasts should be used or dimorphic fungi. Nor is it mentioned that there are at least two papers that evaluate the in vivo capacity of farnesol to treat Candida infections in mouse models.

In Figure 1  the different clinical forms should be described.

In the statistical analysis section, there is no mention of any test or test carried out such as chi-square, student's t test or others.

In Table 2, to facilitate its interpretation, in the optical density column a color scale could be used (as is done with antifungal sensitivity) to define the isolates according to their ability to form biofilms (weak, moderate and strong).

In section 3.2 it is not mentioned at what optical density it is measured and the formula used to calculate the biofilm inhibition should be defined in the material and methods section; as well as how to perform logistic regression.

Both in the results sections and in the conclusions it is mentioned that the greatest fungicidal effect was observed at concentrations between 200-25 microM, which produces a decrease in optical density of between 71-54%, but why is the concentration of 12.5 microM not included which decreases the optical density by 55%?